

# Tropospheric dry layers in the Tropical Western Pacific: Comparisons of GPS radio occultation with multiple data sets

Therese Rieckh[1,2], Richard Anthes[1], William Randel[3], Shu-Peng Ho[1], and Ulrich Foelsche[4,2]

[1]COSMIC Program Office, University Corporation for Atmospheric Research, Boulder, Colorado, USA
[2]Wegener Center for Climate and Global Change, University of Graz, Graz, Austria
[3]National Center for Atmospheric Research, Boulder, Colorado, USA
[4]Institute for Geophysics, Astrophysics, and Meteorology/Institute of Physics, University of Graz, Austria

*Correspondence to:* Therese Rieckh (rieckh@ucar.edu)

**Abstract.** We use GPS Radio Occultation (RO) data to investigate the structure and temporal behavior of extremely dry, high-ozone tropospheric air in the Tropical Western Pacific during the six-week period of the CONTRAST (CONvective TRansport of Active Species in the Tropics) experiment (January and February 2014). Our analyses are aimed at testing if the RO method is capable of detecting these extremely dry layers, and evaluating comparisons with in situ measurements, satellite observations, and model analyses. We use multiple data sources as comparisons, including CONTRAST research aircraft profiles, radiosonde profiles, AIRS (Atmospheric Infrared Sounder) satellite retrievals, and profiles extracted from the ERA (ERA-Interim Reanalysis) and the GFS (U.S. National Weather Service Global Forecast System) analyses, as well as MTSAT-2 satellite images. The independent and complementary radiosonde, aircraft, and RO data provide high vertical resolution observations of the dry layers. However, they all have limitations. The coverage of the radiosonde data is limited by having only a single station in this oceanic region; the aircraft data are limited in their temporal and spatial coverage; and the RO data are limited in their number and horizontal resolution over this period. However, nearby observations from the three types of data are highly consistent with each other and with the lower-vertical resolution AIRS profiles. They are also consistent with the ERA and GFS data. We show that the RO data, used here for the first time to study this phenomenon, contribute significant information on the water vapor content and are capable of detecting layers in the tropics and subtropics with extremely low humidity (less than 10 %), independent of the retrieval used to extract moisture information. Our results also verify the quality of the ERA data set, giving confidence to the reanalysis and its use in diagnosing the full four-dimensional structure of the dry layers.

## 1 Introduction

Water vapor is the most important green house gas in the troposphere, yet it is still the parameter with the highest uncertainty in weather and climate models. All current humidity observation techniques have limitations for analyzing global water vapor fields. For example, nadir-viewing satellite instruments (e.g. infrared (IR) or microwave (MW) sensors in space) are restricted by their low vertical resolution. IR sounders cannot observe under clouds. Radiosonde coverage is sparse or non-existent over the open oceans. The Radio Occultation (RO) method does not suffer from these limitations, but water vapor information can



only be derived by using a combination of RO data and information on temperature from another source (observations or model). In addition, RO observations have significant errors in the lower tropical troposphere if super-refraction conditions exist (Sokolovskiy, 2003).

A number of studies have shown good agreement between RO and radiosonde moisture profiles (e.g., Kishore et al., 2011; Ho et al., 2010). However, these comparisons are limited to locations over land, and typically do not include extremely dry air (air with relative humidities less than 10 %). In this paper we study the ability of RO to measure extremely dry air in the tropics and subtropics, using for comparison high-resolution aircraft profiles, radiosondes, IR and MW data, and the ERA (Berrisford et al., 2011).

Dry regions of the tropical and subtropical lower and mid troposphere have a strong radiative impact on the climate system through their ability to radiate heat to space, preventing a "runaway greenhouse effect" (Pierrehumbert, 1995). Furthermore, they suppress deep convection (Brown and Zhang, 1997), are connected to cumulus congestus cloud top heights (Johnson et al., 1996), and affect boundary layer height and humidity (Parsons et al., 2000).

A number of studies have addressed the so-called dry intrusions in the normally moist mid-and lower troposphere of the Tropical Western Pacific. They were first investigated during TOGA-COARE, the Coupled Ocean-Atmosphere Response Experiment of the Tropical Ocean and Global Atmosphere project (Webster and Lukas, 1992). Mapes and Zuidema (1996), using soundings from TOGA-COARE, found that dry layers are generally too dry and not warm enough to be interpreted as adiabatic displacements within the tropics. Instead, they suggest subtropical origin. Dry layers have strong horizontal and vertical moisture gradients and sharp temperature inversions at the lower edge. They are stabilized by extraordinary cooling of the underlying moist air and heating of the dry air layer, thus preventing convection.

Cau et al. (2005) investigated the radiative impact and origin of dry intrusions observed by radiosonde profiles in the Tropical Western Pacific using 40-year European Centre for Medium-Range Weather Forecasts Reanalysis (ERA-40) wind and humidity data. They showed an outgoing longwave radiation increase of $3\,\mathrm{W\,m^{-2}}$ per $100\,\mathrm{hPa}$ for dry intrusions with relative humidities of less than 20 %, almost independent of altitude. They pointed out the importance of the humidity distribution in a climate change scenario, considering that outgoing longwave radiation is more sensitive to small humidity perturbations in dry environments than in moist regions. Cau et al. (2005) also pointed out that cloud occurrence above or below the dry intrusion reduces the radiative impact. Furthermore, they used back trajectories to show that most dry events were associated with air descending from the extratropics. In their follow-up study, Cau et al. (2006) did a detailed study on the origins of dry air in the tropics and subtropics using trajectory simulations for ERA-40 data for January 1993. They found four major transport mechanisms: (1) the descending branch of extratropical baroclinic waves; (2) the equatorial flank of the jet around subtropical anticyclones; (3) transport at regions of minimum subtropical jet strength via equatorward descent across the jet exit; and (4) dry air centering in the upper troposphere between regions of deep convection (see also Fig. 9 in Cau et al. (2006)). Regarding dry layer occurrence, Casey et al. (2009) created a five-year climatology on dry layers between $600\,\mathrm{hPa}$ and $400\,\mathrm{hPa}$ over deep convective regions of the tropical oceans using AIRS (Atmospheric Infrared Sounder) data. Their results show large spatial and seasonal variability for different ocean basins, pointing out the limits of applying case study trends to the whole basin.





Finally, Randel et al. (2016) performed a detailed comparison between aircraft measurements from the CONvective TRansport of Active Species in the Tropics (CONTRAST) experiment and GFS meteorological analyses, demonstrating that the analyses accurately capture the behavior of subtropical dry layers. A global climatology from GFS data show that the dry layers are a ubiquitous feature of the subtropics, with maximum occurrence frequency in the winter hemisphere (linked to the strongest subtropical jets). The subtropical dry layers are highly correlated with enhanced ozone in both hemispheres, arguing for a source in the extratropical upper troposphere – lower stratosphere (UTLS).

A number of studies confirmed the capability of RO measurements to monitor the dry atmosphere (above around 8 km) (Foelsche et al., 2008, 2009) and for climate change detection (Leroy et al., 2006; Ho et al., 2009; Steiner et al., 2011; Ho et al., 2012). RO data feature inherent high accuracy and precision, high vertical resolution (100 m to 200 m), all-weather capability, and long-term stability (Anthes, 2011), making them highly valuable for studying a large number of atmospheric phenomena. Vergados et al. (2015) studied the spatial variability of relative humidity ($RH$) from RO compared to ECMWF (European Centre for Medium-Range Weather Forecasts) and MERRA (Modern-Era Retrospective analysis for Research and Applications), focusing on time–average seasonal behavior; these comparisons suggest an overall reliable behavior of RO-derived humidity fields. So far no study has focused on RO and extreme dryness ($RH < 10\,\%$).

$RH$ is computed from measured water vapor pressure and saturation water vapor pressure over liquid or ice, depending on the temperature. The liquid formulation is used according to Murphy and Koop (2005).

This paper is structured as follows: In Section 2 we summarize the RO technique, the CONTRAST field campaign, and all other data sets we used. In Section 3 we show some example profile comparisons, explain features of dry layers, and discuss the contributions of a-priori (first guess) data and RO observations in the 1D-Var. Section 4 focuses on one specific case in detail. In Section 5, we give a short overview of the results using all collocation pairs available. Section 6 discusses the global occurrence of dry layers derived from RO data. Section 7 provides a summary and conclusions.

## 2   Data and methods

### 2.1   The RO Method

The RO method (Melbourne et al., 1994; Hajj et al., 2002; Kuo et al., 2004) is a limb-sounding technique that provides near-vertical profiles of atmospheric refractivity $N$. The relation of $N$ to atmospheric temperature $T$, pressure $p$, and water vapor pressure $e$ can be approximated by the Smith and Weintraub (1953) formula:

$$N = 77.6\frac{p}{T} + 3.73{\times}10^5\,\frac{e}{T^2} + [...] \tag{1}$$

Additional terms accounting for contributions from liquid water and the ionosphere can be neglected or are accounted for earlier in the retrieval. In the so-called dry air retrieval, the "dry temperature" is computed using Eq. 1 under the assumption $e = 0$. For a detailed retrieval description, see Kursinski et al. (1997).





Water vapor pressure $e$ in Eq. 1 cannot be determined from an observed $N$ without ancillary temperature data from some other source (either observations, a model, or analysis). The two common techniques for this calculation are discussed in Appendix A, as well the influence of the ancillary data in the One-Dimensional Variational (1D-Var) Retrieval (Appendix B).

For this study, we downloaded data from CDAAC [1] (COSMIC Data Analysis and Archive Center) for the RO missions

COSMIC (Constellation Observing System for Meteorology, Ionosphere and Climate; reprocessed data cosmic2013), GRACE (Gravity Recovery and Climate Experiment; postprocessed), METOP–A (Meteorological Operational Polar Satellite – A; reprocessed data metopa2016), METOP–B (Meteorological Operational Polar Satellite – B; postprocessed), and TerraSarX (postprocessed).

CDAAC provides profiles of physical parameters, which are derived by using a 1D-Var retrieval (COSMIC, 2005)). In the

1D-Var, ERA profiles (interpolated to the location and time of the RO measurement) are used as the initial (first guess or a-priori) temperature and moisture profiles in the iteration procedure. Furthermore, CDAAC also provides these a-priori profiles, and collocated profiles from other (re-)analyses. In this study we use these RO-collocated profiles from ERA and GFS for comparisons.

## 2.2    The CONTRAST Experiment

The CONvective TRansport of Active Species in the Tropics (CONTRAST) Experiment was conducted over the Western Pacific warm pool region during the season characterized by intense convective storms to study the impact of deep convection on chemical composition and ozone photochemical budget (Pan et al., 2016). The experiment was conducted from Guam (13.5° N, 144.8° E) using the NSF/NCAR Gulfstream V (GV) research aircraft during January and February 2014. During the campaign, 16 research flights were conducted. Most research flights included several vertical profiles (covering altitudes

from 0.1 km to 15.2 km), and together with take-offs and landings at Guam, there were over 80 vertical profiles obtained during the experiment. We use the aircraft observations of temperature, pressure, and water vapor pressure to derive high-resolution vertical profiles of $RH$.

## 2.3    ERA-Interim Reanalysis

In addition to the RO-collocated ERA profiles (as described in 2.1), we downloaded ERA-Interim Reanalysis fields from

European Centre for Medium-Range Weather Forecasts (2009) for the time range of the CONTRAST experiment. They are available every six hours at 00, 06, 12, and 18 UT. We use the data on the lowest 27 levels, from 1000 hPa to 100 hPa. ERA uses a 4D-Var method and assimilates radiosonde humidities, AIRS radiances, and RO bending angles.

---

[1] http://cdaac-www.cosmic.ucar.edu/cdaac/





## 2.4 Radiosonde, AIRS, and MTSAT-2 Observations

Radiosonde (RS) data from Guam were downloaded from NOAA [2]. Data are available at approximately midnight and noon UT (9am and 9pm local times, respectively). Measurements are taken at standard pressure levels and significant thermodynamic levels. To convert the pressure grid to altitude, we used a constant temperature gradient of $6.5\,\mathrm{K\,km^{-1}}$.

5      AIRS is a cross-track scanning instrument on the NASA Aqua satellite. Its sun-synchronous, near-polar orbit is designed to cross the equator from south to north at 13:30 local time. The NASA Goddard Sciences Data and Information Center provides AIRS retrieved data products, such as profiles of physical parameters (temperature, humidity) and trace gas constituents on a daily basis. We downloaded the version 6 Level 2 standard Retrieval data [3].

     MTSAT-2 is a geostationary satellite located over Australia, East Asia, and the Western Pacific, operated by the Japan 10   Meteorological Agency. Detailed information can be found at Knapp (2008). MTSAT-2 carries an imaging telescope, backed by detectors for five wavelength channels.

     We use data from the infrared channel ($10.3\,\mu\mathrm{m}$ to $11.3\,\mu\mathrm{m}$) and the water vapor channel ($6.5\,\mu\mathrm{m}$ to $7\,\mu\mathrm{m}$). We downloaded MTSAT-2 data from NOAA [4].

     The sampling characteristics of the different observation and model data sets compared in this study vary greatly. The 15   radiosonde and aircraft data are essentially point measurements. Radiosonde measurements are taken on standard pressure levels and significant thermodynamic levels, which results in the vertical resolution varying strongly within the profile (from less than $20\,\mathrm{m}$ to almost $1000\,\mathrm{m}$). The vertical resolution of the aircraft measurements is around $10\,\mathrm{m}$. The horizontal footprint of the radio occultation profiles is $\sim$$200\,\mathrm{km}$ while the vertical resolution is $100\,\mathrm{m}$ to $300\,\mathrm{m}$ (Anthes, 2011). According to the AIRS Instrument Guide [5], AIRS has a horizontal footprint of $13.5\,\mathrm{km}$ at nadir, increasing to $41\,\mathrm{km}\times21.4\,\mathrm{km}$ at the limits of 20   its scan, and a vertical resolution of $\sim$$1\,\mathrm{km}$ for temperature and $\sim$$2\,\mathrm{km}$ for humidity. For the two models used in this study, we used GFS and ERA interpolated to the time and location of the RO profile. Furthermore, we used the ERA field, for which the horizontal footprint is given by the horizontal grid size ($0.7°\times0.7°$, about $78\,\mathrm{km}\times(68\text{ to }78)\,\mathrm{km}$, depending on the latitude). The vertical resolution is given by the model pressure levels every $25\,\mathrm{hPa}$ or $50\,\mathrm{hPa}$ (resulting in a vertical resolution between $200\,\mathrm{m}$ to $1000\,\mathrm{m}$).

25      Because both radiosondes and the aircraft measurements are essentially rapid-response point values and have high vertical resolution, they are capable of measuring turbulence and small-scale horizontal features (such as individual clouds). As such, they may contain large sampling errors when compared to the much larger scale of observations associated with AIRS, RO and the models, especially when measuring fields with high temporal and spatial variability such as water vapor and relative humidity.

30      In addition, all the observations and model data occur at different locations and times. The radiosonde, RO and aircraft observations occur at different horizontal positions in the vertical (can be $\sim$$100\,\mathrm{km}$ to $200\,\mathrm{km}$) as the balloons, satellites and

---

[2]https://www.ncdc.noaa.gov/data-access/weather-balloon/integrated-global-radiosonde-archive

[3]ftp://airsl2.gesdisc.eosdis.nasa.gov/ftp/data/s4pa/Aqua_AIRS_Level2/AIRS2RET.006/

[4]http://www.ncdc.noaa.gov/gibbs/availability/2014-02-20

[5]http://disc.sci.gsfc.nasa.gov/AIRS/documentation/airs_instrument_guide.shtml





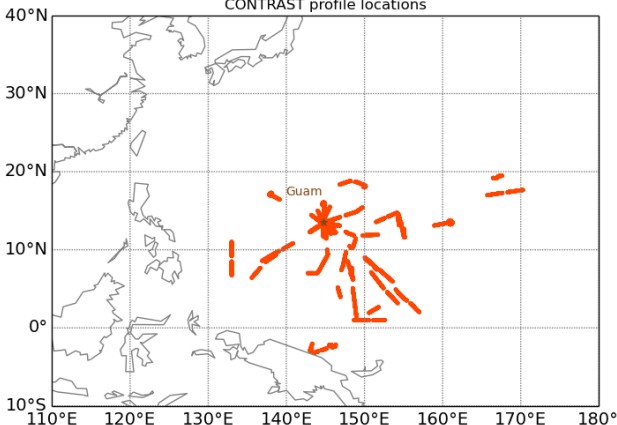

**Figure 1.** Profiles extracted from the CONTRAST flights during the experiment

aircraft move during the "vertical" sounding. All of these differences make comparisons challenging, adding to the errors associated with each individual data set, and must be considered when interpreting the results.

## 2.5 Collocating CONTRAST and RO Profiles

From the 16 research flights, we extracted 75 profiles that extend over at least 6 km altitude, and are within the region $5°$ S to $20°$ N latitude, $130°$ E to $170°$ E longitude (Fig. 1). We tried different criteria for maximum time and distance for collocating RO with CONTRAST profiles. We made these comparisons only for the lowest 10 km, since RO observations do not provide reliable moisture values at altitudes above about 8 km (Kishore et al., 2011). Aircraft-RO profile pairs with less than 4 km overlap in the vertical were discarded (CONTRAST profiles are mostly limited by their maximum altitude, RO profiles are limited by their minimum altitude). The time and space coincidence criteria tested included 3 h and 600 km; 12 h and 300 km; and 24 h and 200 km, yielding 37, 41, and 24 profile pairs, respectively. As expected, we found that the agreement was better for the closest (in time and space) profiles (not shown).

## 3 Individual Profile Comparisons

Fig. 2 shows an example of a dry layer sampled by the CONTRAST Research flight 2 profile d (rf02d), and the collocated RO profile (METOP–A) for the parameters $RH$, $T$, $q$, and $N$. $RH$ (upper left) shows a typical dry layer structure. There is a strong drop in $RH$ at the bottom of the layer (at $\sim$2.5 km) from $> 80\%$ to $< 10\%$. All profiles (CONTRAST, RO, ERA, and GFS) show extremely dry air above 4 km. This layer is particularly thick and the $RH$ remains below 20 % up to 10 km. The RO profile shows a remarkably similar shape when compared to CONTRAST, including the sharp humidity gradient at the bottom of the dry layer. Both ERA and GFS show a less sharp vertical moisture gradient, partly due to lower vertical resolution. Profile differences are largest in the lowest 2.5 km, where RO and ERA $RH$ are up to 20 % lower than the CONTRAST and GFS $RH$.



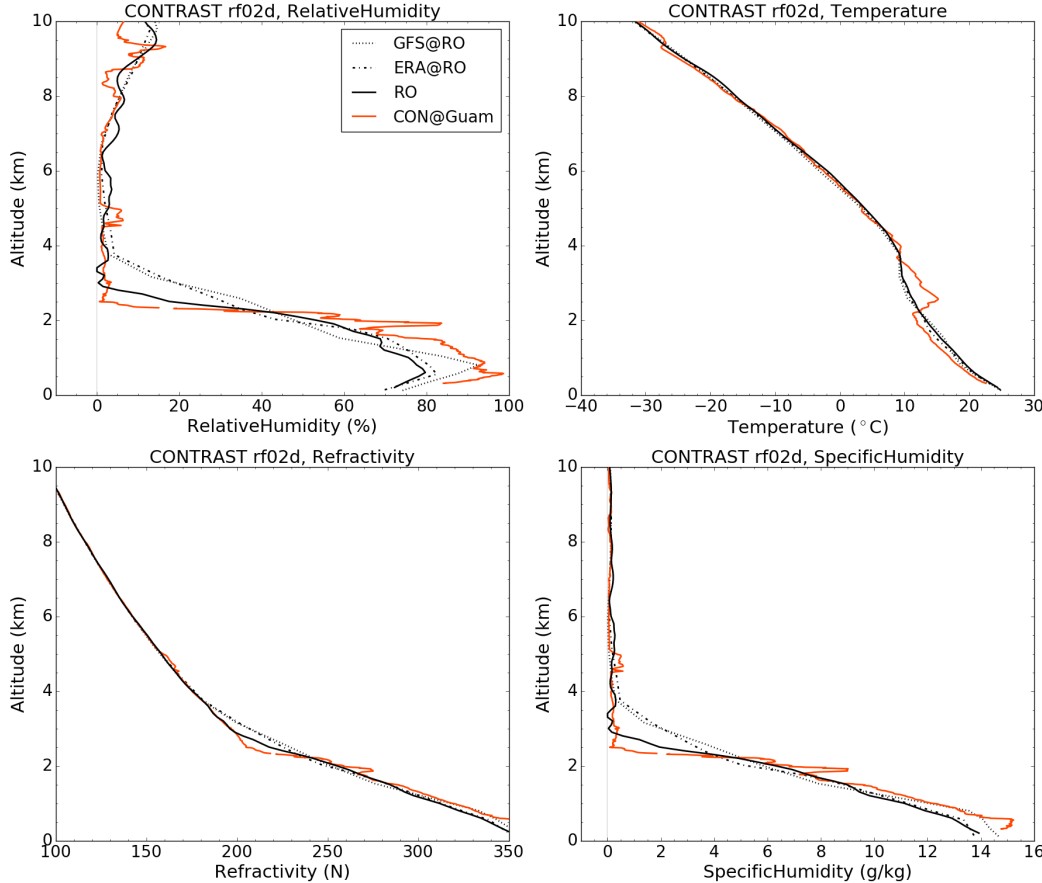

**Figure 2.** $RH$, $T$, $N$, and $q$ profiles for CONTRAST (solid orange), RO (solid black), and RO-collocated profiles (ERA: dashed-dotted black; GFS: dotted black). Profile times and locations: CONTRAST: 2014-01-14, 00:58–01:11 UT at 19.2° N, 166.5° E; RO 2014-01-13, 22:18 UT at 17.7° N, 164.4° E. The profiles are about 3 h and 390 km apart.

The aircraft $T$ profile (Fig. 2, upper right) shows a strong inversion at the altitude of the bottom of the dry layer, as has been described by, e.g., Mapes and Zuidema (1996). Both ERA and GFS as well as RO do not detect this strong inversion. RO generally has the capability to resolve such strong inversions in the middle and upper troposphere (Anthes, 2011). We conclude that RO not showing the $T$ inversion could be explained by two factors: 1) CONTRAST and RO have different $N$ values at this altitude (see Fig. 2, lower left). This implies that there has to be a difference in $T$ and/or $e$ at this altitude, too. 2) The 1D-Var retrieval generally produces an RO $T$ close to the first guess $T$, which does not show the inversion, and changes mainly $e$ in the adjustment of the first guess $N$ towards the measured RO $N$.

Specific humidity $q$ (lower right) shows extremely dry conditions above 2.5 km, but in less extreme cases dry layers are harder to detect using these parameters since they generally decrease exponentially with altitude. Thus we mainly use $RH$ to investigate dry layers.





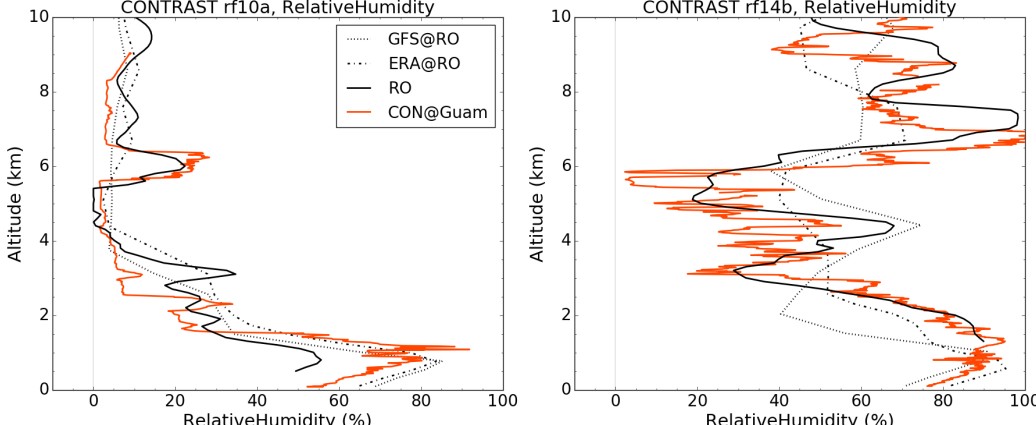

**Figure 3.** $RH$ profile for CONTRAST (solid orange), RO (solid black), and RO-collocated profiles (ERA: dashed-dotted black; GFS: dotted black). Profile times and locations: CONTRAST (left): 2014-02-08, 00:17–00:32 UT at $13.5°$ N, $144.8°$ E; RO (left): 2014-02-07, 22:57 UT at $13.9°$ N, $148.1°$ E. CONTRAST (right): 2014-02-22, 09:09–09:35 UT at $13.5°$ N, $144.8°$ E; RO (right): 2014-02-22, 12:06 UT at $15.6°$ N, $148.1°$ E.

Fig. 3 shows two more examples for CONTRAST-RO pairs (METOP–B and COSMIC-FM6). The left panel depicts a dry layer, extending from about 2.5 km upwards for CONTRAST. The RO $RH$ shows a very similar overall structure. Major differences are again in the lowest 2 km, where CONTRAST, GFS, and ERA $RH$ are up to 20 % higher than RO $RH$, and between 2.5 km and 4 km, where CONTRAST $RH$ is up to 20 % lower. Furthermore, both GFS and ERA miss the 1 km thick

moist layer around 6 km.

Fig. 3, right, shows profiles with two drier layers in the mid troposphere (at 3 km and 5 km), but no extreme dryness. Both ERA and GFS show the correct overall shape of the $RH$ profile, but they are often 20 % to 30 % drier or moister than the CONTRAST $RH$. Again, RO captures more vertical structure than the models.

Generally both of these examples show how well the RO $RH$ profile structure agrees with the one from CONTRAST.

**4   Case study: Research flight 13**

We found that many profile pairs matched very closely, however, some of the pairs showed $RH$ differences of more than 60 % at certain levels. In this section we look into one of these cases in more detail to help explain these strong discrepancies. We consider two specific profiles, one measured by CONTRAST starting at Guam on 2014-02-19, 17:00 UT (Fig. 4, top), and one landing at Guam a few hours later (2014-02-20, 00:22 UT) (Fig. 4, bottom).

For both cases, the left side shows the profiles for CONTRAST, a collocated RO, the RO-collocated ERA and GFS profiles, a RS launched at Guam ($RS_G$), and two AIRS profiles (one closest to Guam, labeled as $AIRS_G$; and one closest to the RO profile, labeled as $AIRS_{RO}$). Table 1 lists the time differences and distances between these profiles.



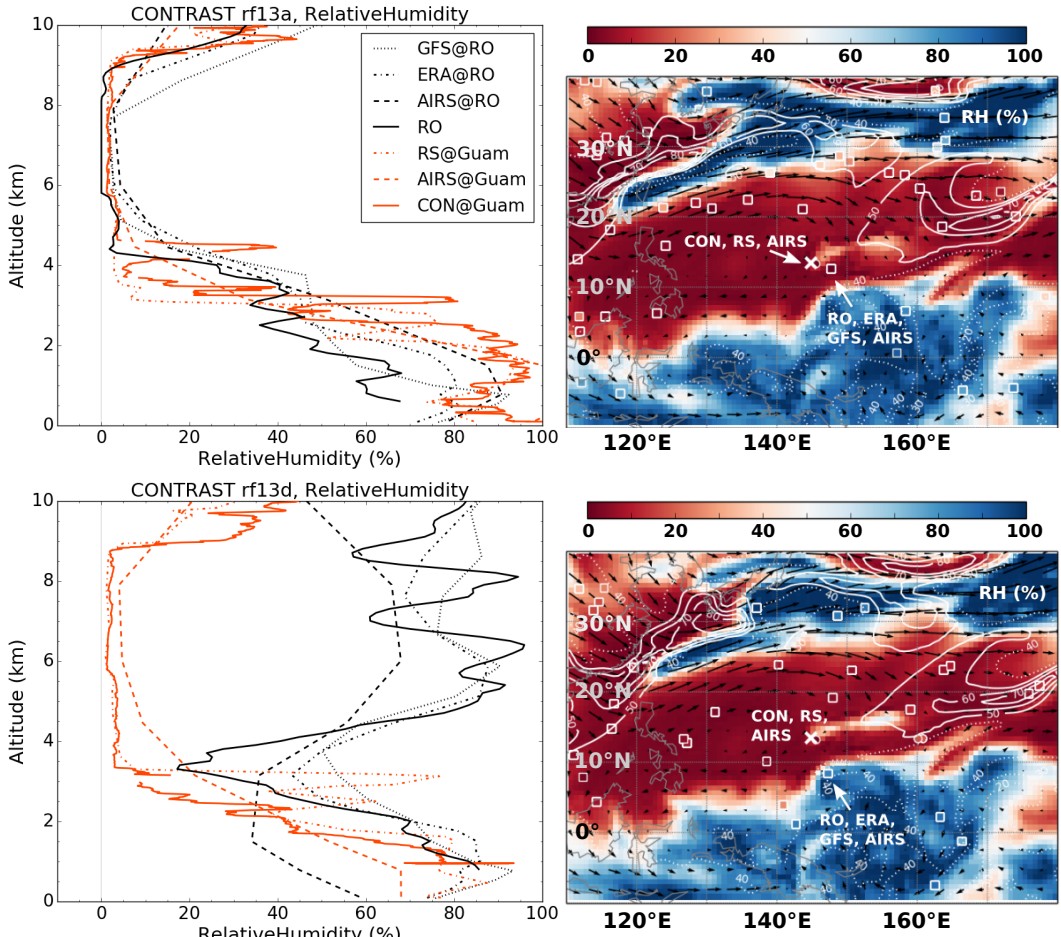

**Figure 4.** Two snapshots of the troposphere for (top) 2014-02-19 around 18:00 UT, and (bottom) 2014-02-20 around 00:00 UT. Left panels: profiles for CONTRAST, RO, ERA, GFS, RS, and AIRS for the two days. Right panels: ERA $RH$ (%, color shading) at 500 hPa, ozone values (white contours: values $\geq$ 50 ppbv are solid, otherwise dashed), and winds (black arrows). The white X marks the location of Guam. The locations of CONTRAST, RO, RS, and AIRS profiles are marked. Additional squares indicate other RO measurements. The squares marking the locations of the RO and CONTRAST locations are colored according to the $RH$ of each profile at the 500 hPa level (matching the color bar).





**Table 1.** Distances and time differences between different the collocated profiles of rf13a and rf13d at the lowest point for each profile. Note that both the aircraft and RS profile will move away from Guam with higher altitudes, and that RO profiles are also not completely vertical, especially in the lower and mid troposphere.

| **Profile 13a** | | | | | |
|---|---|---|---|---|---|
| | RO–CON | $RS_G$–CON | $AIRS_G$–CON | $AIRS_{RO}$–CON | $AIRS_{RO}$–RO |
| Distance (km) | 345 | 4 | 16 | 331 | 18 |
| Time Diff (min) | -117 | -289 | -70 | -70 | 47 |
| **Profile 13d** | | | | | |
| | RO–CON | $RS_G$–CON | $AIRS_G$–CON | $AIRS_{RO}$–CON | $AIRS_{RO}$–RO |
| Distance (km) | 599 | 4 | 34 | 588 | 128 |
| Time Diff (min) | 87 | 1 | 236 | 234 | 147 |

The right plots shows the ERA $RH$ field at 500 hPa, closest in time to the respective CONTRAST profiles. Also shown are the ERA ozone values and winds. The location of Guam is marked by a white X. CONTRAST, RO, RS, and AIRS profiles are labeled. Additional squares indicate other RO measurements. The color filling of the RO symbols (white squares) varies with the $RH$ of the RO observation at this level, with the same color code as the ERA $RH$ analysis (color bar). It is noteworthy that in almost all the cases the colors (and hence $RH$) agree very closely.

In Fig. 4, top left, all profiles show a deep dry layer. The depth of the layer varies slightly between the data sets. For CONTRAST, RS, and RO the depth with $RH \leq 10\%$ varies between 5 km and 6.5 km. Both the ERA and GFS profiles and both AIRS profiles show a less sharp transition from moist to dry (partly caused by lower vertical resolution). The models also show a generally thinner dry layer. Major differences between the data sets occur below 2 km, where RO is significantly drier than all other data sets. The GFS profile agrees with the drier RO profile down to about 1 km and then strongly increases in $RH$. Sometimes super-refraction can cause a dry bias in RO profiles in the lowest few kilometers, however, super refraction does not occur in this particular profile ($\mathrm{d}N/\mathrm{d}z < -157$ N-units km$^{-1}$, see Sokolovskiy (2003)), as the minimum value of $\mathrm{d}N/\mathrm{d}z$ computed from the RO profile is only $-103$ N-units km$^{-1}$.

Fig. 4, top right, shows the ERA $RH$ field at 500 hPa for the whole region. It shows the high horizontal variability of moisture. In some areas, extremely strong horizontal $RH$ gradients occur, which clearly mark the edge of the dry air mass. In these areas $RH$ can increase from less than 20 % to more than 70 % within 100 km. And since this is a model field, in which the gradients are likely to be smoother than the real atmosphere, the actual gradient could be even sharper. In this figure, it is also clearly visible that all the profiles from the left panel are located in the same air mass.

Next we consider the same region about 6 hours later (Fig. 4, bottom panels). The profiles (left) show that the dry layer persists at Guam and is even deeper for CONTRAST. The RS profile and the AIRS profile at Guam still show a very deep dry layer. However, the RO shows only a very shallow dry layer, and RH increases from less than 20 % to more than 80 % between 3.5 km and 4.5 km. The RO-collocated ERA and GFS profiles lack the dry layer entirely, having their $RH$ minima at 45 %





and 50 %, respectively. They also both increase above 3.5 km to around 80 %. The AIRS profile at the RO location starts much drier than RO, ERA, or GFS at the surface. Overall, the AIRS profile has the same shape as RO, ERA, and GFS. It has weaker dry-moist transitions, similar to the (re-)analyses, and it stays drier than the other data sets above 4 km. Comparing the two AIRS profiles confirms the credibility of both RO and ERA: the strong difference between the aircraft profile and RO profile

is neither an ERA nor an RO error, but caused by the combination of an imperfect collocation and strong spatial variability.

The lower right panel in Fig. 4 shows that the RO (and thus also the RO-collocated ERA and GFS) profiles are located in the moist air mass, just a few tens of km from the edge of the moist-dry boundary, with much higher moisture values above 3 km. This explains the very different profiles in the left panel.

The ERA $RH$ field also shows the large extent of extremely dry air, such as the dry band between $\sim 10°$ N and $20°$ N in

Fig. 4. Previous studies on dry intrusions described them often as narrow "dry tongues" (300 km width), however, the ERA field shows a width of 1600 km to 2200 km.

Finally, Fig. 5 shows satellite images from MTSAT-2 for approximately the same time as the ERA field in Fig. 4, bottom right. The satellite images are cropped to the latitude and longitude range as ERA (as much as the different projections allowed). Guam and the RO are marked by an X.

The left panel depicts the brightness temperatures from infrared in an atmospheric window (IR, at 10.3 $\mu$m to 11.3 $\mu$m). It is derived from terrestrial IR radiation emitted by the Earth, cloud tops, and the atmosphere. Color enhancement shows the high, cold cloud tops south of Guam. Conditions are clear around Guam and the CONTRAST profile, but the RO profile is located in cloudy air.

The right panel shows the image from the MTSAT-2 water vapor (WV) channel. It depicts brightness temperatures derived

from the water vapor emission spectrum between 6.5 $\mu$m and 7 $\mu$m. Higher amounts of water vapor absorb more radiation, which is re-emitted. Thus regions with high amount of water vapor, especially in the upper troposphere or above clouds, will have a higher brightness temperature. When compared to ERA (Fig. 4, bottom right), we see that the low $RH$ region has a much lower brightness temperature than the moist region south of Guam.

## 5   Results of all Collocations

To get a general overview of profile pair differences we computed statistics (not shown) and created scatter plots for all collocated profiles. Since humidity has a strong spatial variation (as seen in Fig. 4, right) and also varies strongly with time, we used different sets of collocation criteria (described in Section 2.5. CONTRAST profiles are smoothed by taking 60 s averages in each profile (resulting in about 300 m–500 m vertical resolution). The RO profiles are interpolated to the related CONTRAST grid. We did comparisons for RO and CONTRAST, ERA and CONTRAST, and GFS and CONTRAST. Generally, all 3 sets

of collocation criteria and all data set comparisons show similar results for both the profile statistics and scatter plots. Thus we only show plots for the 3 h and 600 km criteria in the following.

Fig. 6 shows the scatter plots for refractivity $N$, temperature $T$, specific humidity $q$, and relative humidity $RH$ for RO compared to CONTRAST. The color of the dot indicates the altitude for 0 km to 10 km as shown in the colorbar. The black



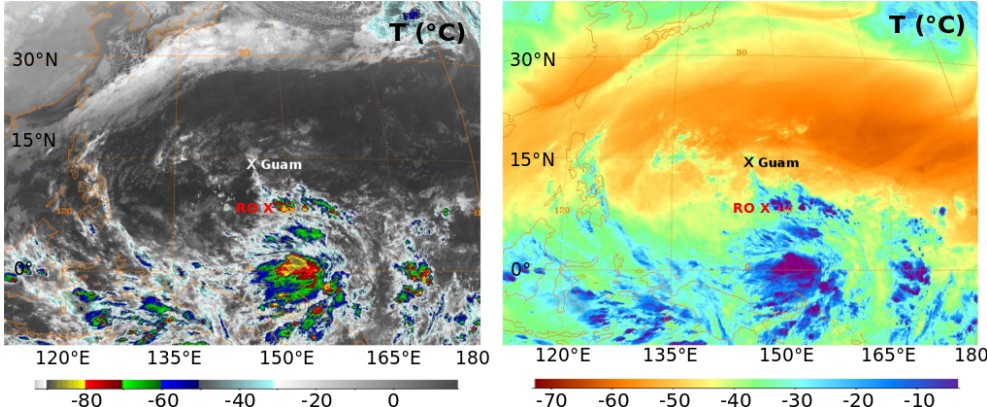

**Figure 5.** MTSAT-2 satellite images of brightness temperatures for 2014-02-20 around 00 UT. Left: MTSAT-2 IR; right: MTSAT-2 IR water vapor.

solid line is the fitted linear regression, the dashed black line indicates perfect agreement between the two data sets (slope=1, intercept=0).

The upper left panel depicts the $N$ comparison between RO and CONTRAST. $N$ decreases exponentially with altitude, so the spread is larger at lower altitudes (blue) than at higher altitudes (red). Overall, there is a high correlation between the two data sets, and the fitted regression (solid black line) agrees well with the line of perfect agreement (dashed black line).

$T$ (top right) shows very good agreement (high correlation and little spread). We found a small warm bias of 0.5 K to 1.5 K in CONTRAST temperatures when compared to any of the other data sets (RO, ERA, GFS). To test how much influence the collocation criteria have, we interpolated the ERA field to the CONTRAST profile location (spatial difference for latitude and longitude $< 0.5°$, time difference less than 30 minutes), which yielded a slightly smaller, more uniform bias. We conclude that there is likely a small $T$ bias in the aircraft temperatures, possibly because of the effects of solar radiation as most of the flights occurred during the daytime.

The specific humidity $q$ is depicted in the bottom left panel. The spread appears larger for low values (very dry air, $< 1\,\mathrm{g\,kg^{-1}}$), but the scales in this panel are logarithmic, which makes small differences of dry values appear larger. RO $q$ values are biased positively compared to CONTRAST for low $q$ values, and biased negatively for high $q$ values. Comparison on $q$ between CONTRAST and GFS in Randel et al. (2016) shows a similar bias.

$RH$ plots (bottom right) are highly scattered and have a lower correlation coefficient of around 0.75, which is a signature of the high variability of $RH$ over short spatial and temporal scales. The large spread can be explained by several factors: 1) $RH$ is sensitive to both small variations in $T$ and $q$, thus sampling differences or errors of both $T$ and $q$ contribute to differences in RH; 2) $RH$ does not have a "typical" vertical profile such as $N$, $T$, $q$ do (with an overall decrease with altitude), thus $RH$ shows much more variability with height; and 3) $RH$ can undergo extremely strong changes in the vertical (80 % $RH$ change over 1 km in Fig. 2), which leads to strong differences between two data sets if they do not capture this jump of $RH$ at exactly the same altitude.



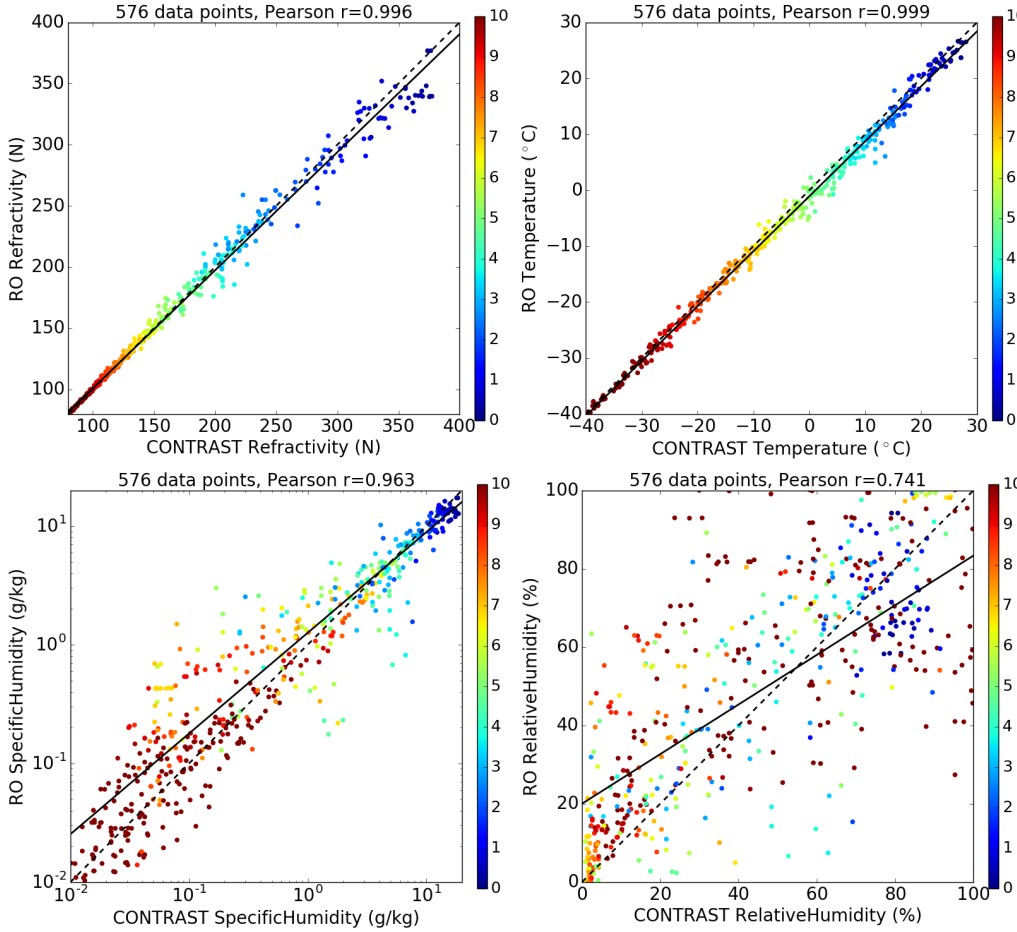

**Figure 6.** Scatter plots for $N$ (top left), $T$ (top right), $q$ (bottom left, logarithmic scale), and $RH$ (bottom right) for all 3 h 600 km collocation pairs for RO and CONTRAST. The color indicates the altitude of the measurement.

Fig. 7, left, compares ERA and CONTRAST $RH$, which shows a similar strong scatter as RO and CONTRAST. This suggests that the high variability in the CONTRAST data set also plays a role in the strong scatter of $RH$. Both the RO and ERA data sets are horizontally smoothed: RO shows an average over about 200 km (limb-sounding), and ERA is interpolated from the nearest grid points to the location of the profile (horizontal resolution is 0.7 deg in latitude and longitude). Finally, Fig. 7, right, compares GFS to ERA (this comparison is done on the ERA pressure grid). The correlation between the two analyses is high, but the scatter is surprisingly large considering these are smooth model data sets. This shows how highly variable $RH$ is.




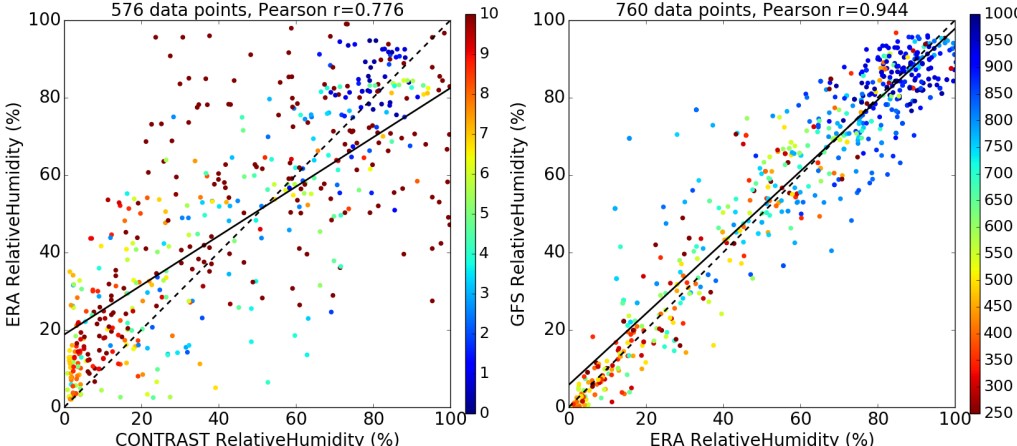

**Figure 7.** Scatter plots for $RH$ for all 3 h 600 km collocation pairs for (left) ERA and CONTRAST on the 60 s smoothed CONTRAST altitude grid; and (right) GFS and ERA on the ERA (pressure) grid. The color indicates the altitude of the measurement (left: km, right: hPa).

## 6 Global Distribution

Having shown that the RO observations are capable of detecting extremely dry layers in the Tropical Western Pacific region, we carried out a global climatology of dry layers using only RO data. AIRS data has been used before to find tropical dry regions within areas of convection (OLR$<$240 W m$^{-2}$) (Casey et al., 2009); however, AIRS cannot provide reliable measurements

below clouds.

Fig. 8 shows the global occurrence (percentage of observations) for $RH < 10\%$ on the 320 K potential temperature level for December-January-February (DJF) and June-July-August (JJA) 2014. We use the 320 K level because dry air travels from the stratosphere into the troposphere along isentropes. The 320 K level is at about 600 hPa or 4.5 km in the tropics, and slopes to higher altitudes ($\sim$9 km) in the extratropics. So the 320 K level represents the mid troposphere in the tropics and the lower

stratosphere in the extratropics.

Fig. 8, left, shows the months DJF. Between 5° N and 25° N, almost the entire latitude band shows an occurrence of dry air for 50 % of the time or more. The only break in the band is off the west coast of North/Central America. In some regions, dry layer occurrence is as high as 75 %, e.g. in parts of the Atlantic or near India and the Arab peninsula. Guam is located just on the edge of the band of high frequency of dry layer occurrence. In the SH, two regions with very strong occurrence are easy to

15 identify: one off the west coast of South America, and one in the southern Atlantic Ocean.

In JJA (Fig. 8, right), dry layers occur over smaller regions of the world, but with a much higher frequency. In the SH, the entire region from mid Atlantic via Africa and the Indian Ocean to Australia shows a frequency of occurrence of 80 % to 100 % of the time. In the NH, only a region in the northern East Pacific and the northeast Africa/eastern Mediterranean region show moderate to high occurrence.



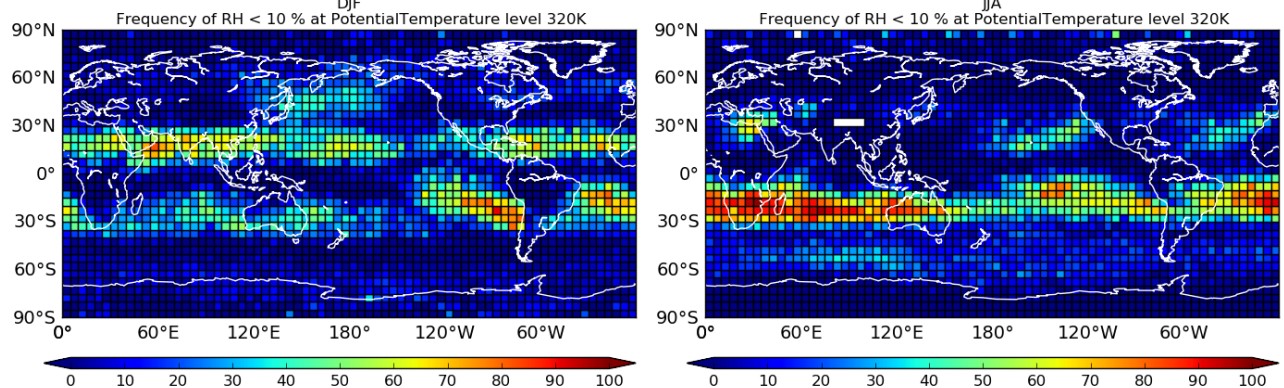

**Figure 8.** Frequency for $RH < 10\%$ at the Potential Temperature level 320 K for DJF and JJA 2014.

Overall, dry layers occur throughout the year in some regions of the SH, but there is also a seasonal cycle with higher occurrence in the winter hemisphere. This behavior is consistent with the dry layer climatology derived from ERA (results not shown), and from GFS data in Randel et al. (2016).

## 7 Conclusions

In this study we compared GPS RO profiles with multiple data sets to examine their ability to detect extremely dry layers in the lower and mid troposphere in the tropics and subtropics. The comparisons were made in the Tropical Western Pacific, making use of the field experiment in that region during January and February 2014. We mainly used data from the NSF/NCAR Gulfstream V research aircraft as a reference for RO profiles. This is a challenging comparison, since the aircraft provides high-resolution profiles consisting of point measurements (capturing a lot of variability), whereas the RO technique is a limb sounding technique (measuring the limb-integrated value at each profile level). Furthermore, we used radiosonde and AIRS profiles, as well as data from the GFS model and ERA reanalysis. Our main findings and conclusions are:

1. Radio occultation is capable of detecting layers with very low humidity in the lower and mid troposphere, despite the fact that the water vapor pressure is only a small fraction of the measured refractivity. Comparing RO to other types of observations also shows that dry layers are captured quite accurately regarding both shape and intensity.

2. Both simple and 1D-Var RO water vapor retrievals yield similar results, showing a profile with fine-scale vertical structure similar to that of the aircraft profile (Appendix A).

3. The 1D-Var retrieval of temperature and water vapor using ERA as a first guess can accommodate a poor first guess in water vapor. Thus the RO water vapor retrieval is not dependent on an accurate first guess. However, a strong difference in $N$ may have an effect of several Kelvin on the retrieved temperature (Appendix B).





4. Both the GFS and ERA analyses show the overall correct structure when compared with CONTRAST aircraft observations and RO. They show less or no small-scale variations or sharp vertical gradients due to a much lower vertical resolution.

5. A case study showed how extremely strong horizontal moisture gradients (more than $50\,\%\ RH$ change within $100\,\text{km}$) can yield profile pairs that strongly disagree, even though they are close in space and time.

6. In a case study in which RO showed much moister air between $4\,\text{km}$ and $8\,\text{km}$ than CONTRAST, even though the two profiles were within $600\,\text{km}$ and 87 minutes of each other, AIRS, the ERA, and the GFS verified the RO sounding. The satellite image showed that the CONTRAST profile was in the dry, clear air, while the RO sounding was a short distance inside the boundary between dry and moist air, confirming that the profiles were located in different air masses.

7. When compared to CONTRAST, RO has a moist bias for low humidity values, and a dry bias for high humidity values.

8. Globally, dry layers occur throughout the year, mainly between 10° and 30° N and S. Occurrence frequency is stronger in the winter hemisphere. The independent RO data confirm both ERA and GFS, which show a very similar seasonal occurrence of dry layers.

In summary, these diverse data sets show generally good agreement, in spite of their large differences in sampling characteristics and technologies. A subsequent paper will use these data sets to describe the four-dimensional variation of atmospheric structure during the CONTRAST period, including the relationship of ozone to the atmospheric dynamics and thermodynamics.

**Appendix A:  RO Water Vapor Retrievals**

The two common techniques to retrieve physical temperature and water vapor profiles from RO are the so-called simple retrieval (Kursinski and Hajj, 2001) and the 1D-Var. For the simple retrieval, $e$ is derived via Eq. 1 using $N_{\text{RO}}$, $p_{\text{model}}$, and $T_{\text{model}}$. Advantages of the simple retrieval are its simplicity and ease of calculation, and its independence of model moisture (and thus independence from errors in model moisture). It provides good results in the lower troposphere if the ancillary temperature data are good. In the 1D-Var procedure, a-priori (first guess) profiles of $T$ and $e$ are obtained from a model or (re-)analysis, and adjusted toward the RO measurements by a statistical optimization procedure (Poli et al., 2002; COSMIC, 2005). The 1D-Var procedure considers the statistics of errors in the RO observations as well as the statistical errors in the a-priori information, to achieve a consistent temperature and water vapor profile that minimizes, in a statistical sense, the errors in $T$ and $e$.

Fig. 9 shows the parameters $RH$, $T$, $q$, and $e$ for the first guess (ERA, solid) and retrieved from RO with the 1D-Var (dashed) and the simple retrieval (dotted) for an example profile.

The $RH$ (top left) is very low between $4\,\text{km}$ and $9\,\text{km}$. The simple retrieval and the 1D-Var agree very well up to $4\,\text{km}$. At $4.2\,\text{km}$, the simple retrieval produces a negative $RH$ (due to negative $e$ values). In the simple retrieval, any error in $T$ will



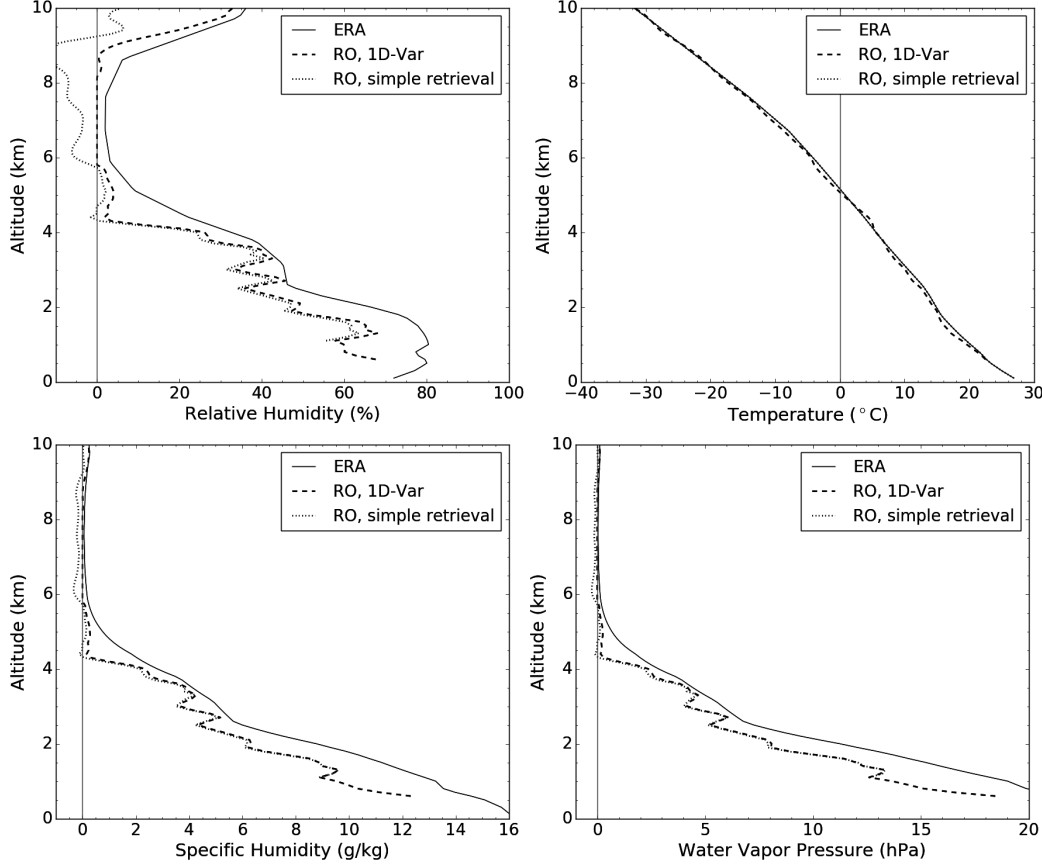

**Figure 9.** $T$, $RH$, $q$, and $e$ profiles for ERA (solid, a-priori), RO 1D-Var (dashed), and RO simple retrieval (dotted).

produce an error in $e$, and for dry air, this error may lead to an unphysical negative value for $e$. The 1D-Var can theoretically also reach negative values in these situations, but it is artificially set to a very small positive value ($10^{-6}$ hPa) in the retrieval.

The RO 1D-Var and first guess (ERA) $T$ (top right) agree very well. (The RO $T$ in the simple retrieval assumes ERA $T$ to be the truth, so it is identical to the ERA $T$ in this figure.)

5    The bottom panels show $q$ and $e$. Both parameters become negative above 4.2 km in the simple retrieval. Generally, these parameters also show that moisture profiles derived from both the simple retrieval and the 1D-Var show much more vertical structure than the ERA profile.

## Appendix B:  Contribution of RO in the 1D-Var

The simple retrieval of water vapor from $N_{\mathrm{RO}}$ and $T_{\mathrm{model}}$ is strongly dependent on an accurate $T_{\mathrm{model}}$ but completely independent

10    from the a-priori model water vapor. The 1D-Var, however, uses the a-priori moisture from ERA. In this section we investigate




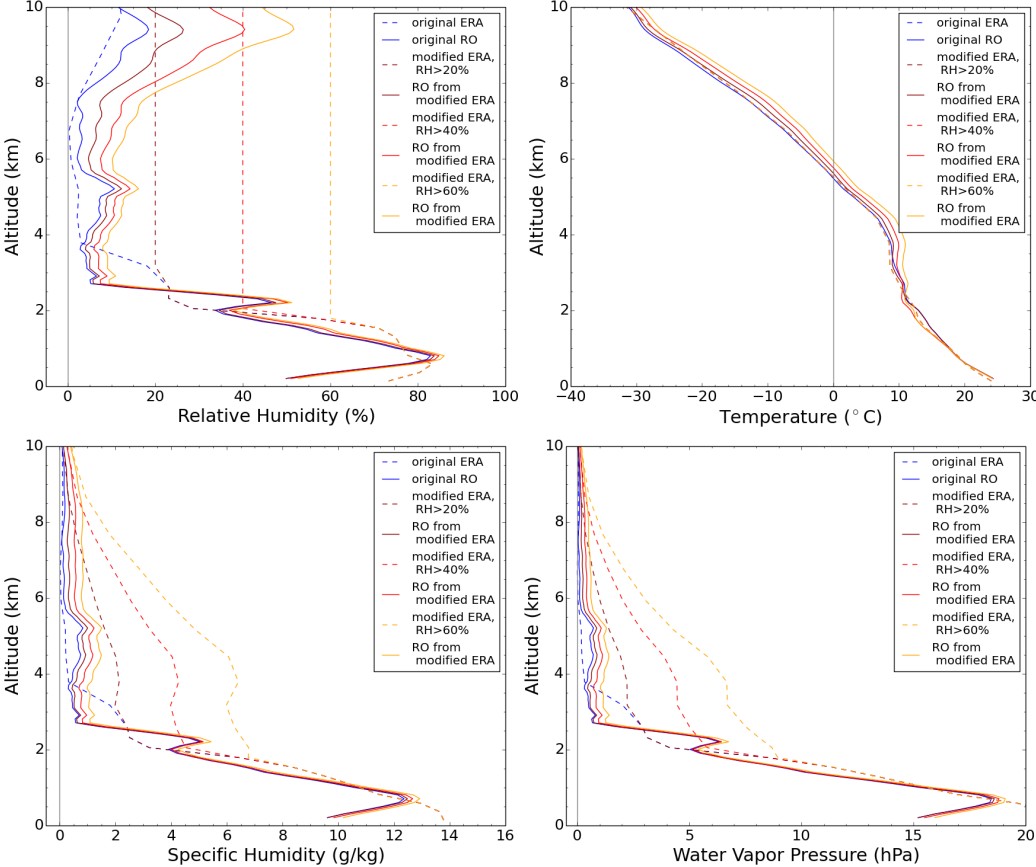

**Figure 10.** $RH$, $T$, $q$, and $e$ profiles with the original ERA first guess (dashed blue), the original 1D-Var output (solid blue), the "high moisture" first guess (dashed, shades of red), and the 1D-Var output for the modified first guess (solid, shades of red).

how the 1D-Var retrieval reacts to a poor humidity first guess and how much information the RO refractivity contributes in the 1D-Var.

As an experiment to test the sensitivity of the 1D-Var method to the first-guess water vapor profile, we change $e$ of the first guess such that the $RH$ is greater than 20 %, 40 %, or 60 % within the lowest 10 km (leaving $T$ unchanged). Then we use these 5   fictitious a-priori "high moisture" data in the 1D-Var retrieval. Fig. 10 shows the results for the original RO and first guess (in blue), and for the "high moisture" cases (in shades of red). The a-priori profiles are dashed and RO profiles are solid.

The top left panel shows $RH$: the three changed a-priori profiles are clearly different from the original profile. At 4 km, the original profile shows a $RH$ of <5 %, while the changed ones are 20 % (dark red), 40 % (red), and 60 % (orange). Using these a-priori in the 1D-Var yields the original (solid blue) RO and the "high moisture" ROs (solid, shades of red). The solid red 10   lines all decrease strongly between 2.5 km and 3 km and follow the shape of the original RO at all levels. Up to 5.5 km, the difference is between 2 % and 6 %. Above that, the $RH$ profiles begin to fan out and differences up to 20 % occur between



5.5 km and 8 km. Overall, the results show that RO refractivity contributes significant information to the water vapor in the 1D-Var retrieval and strongly corrects for the artificially high moisture from the a-priori profiles.

The top right panel in Fig. 10 shows that the RO $T$ profiles show some differences due to the erroneous a-priori water vapor profiles at all levels. Differences are $<1$ K for the original and the 20 % a-priori case, and $\sim$3 K for the original and the 60 % a-priori case.

The lower panels of Fig. 10 depict $q$ and $e$. They show how much the 1D-Var adjusts the high humidity values from the first guess towards the low, realistic humidity values.

*Author contributions.* T. Rieckh and W. Randel formulated the initial idea of this work. R. Anthes and S.-P. Ho helped to develop the design the study further. Together with U. Foelsche, they contributed ideas and provided valuable feedback. T. Rieckh collected the data, performed all the computational work and coding necessary, performed the analysis, and prepared the manuscript. R. Anthes and W. Randel contributed significantly to the data analysis and the writing.

*Acknowledgements.* Rieckh, Anthes, and Ho were supported by the NSF-NASA grant AGS-1522830. Randel was supported through the NSF-UCAR cooperative agreement for the management of NCAR and NASA RNSS Science Team grant NNX12AQ18G.

The CONTRAST data set was provided by NCAR/EOL under sponsorship of the NSF (http://data.eol.ucar.edu/). The ERA data were provided by the Data Support Section of the Computational and Information Systems Laboratory, NCAR, which is sponsored by the NSF. We thank the COSMIC CDAAC team for providing the RO Level-2 data. We also thank Drs. Eric DeWeaver (NSF) and Jack Kaye (NASA) for their long-term advice and support to the COSMIC science program. COSMIC is supported by the National Space Office (NSPO) of Taiwan, NSF, NASA, NOAA, and the U.S. Air Force.



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
