# Peer review of "Tropospheric dry layers in the Tropical Western Pacific: Comparisons of GPS radio occultation with multiple data sets"

_Atmospheric Measurement Techniques, 2016_

## Referee Comment (RC1) · Anonymous Referee #1 · 22 Sep 2016

General comments:

The manuscript presents results from a very interesting intercomparison of observations collected during an experiment in the tropical western pacific with other in-situ or satellite observations and with model data, with a focus on humidity. The presentation is very clear, and the manuscript is written in very good English. Still, a few minor issues should be addressed or answered by the authors before publication.

Specific comments:

Page 2, lines 2-3: In my opinion there is no doubt in the literature that super-refraction conditions do exist. I therefore recommend to reformulate: "if super-refraction conditions exist" -> "under super-refraction conditions"

Section 2.2 ("The CONTRAST experiment"): For the reader's sake, there should be a very brief explanation of the sensors used for the aircraft observations, in particular the derivation of water vapor pressure, as this is a central quantity in this study. The Pan et al. paper has no full reference and could not be found.

Section 2.3 ("ERA-Interim Reanalysis") and section 2.4: The ERA-Interim dataset is available on the original 60 model levels, as well as interpolated to a predefined set of (coarser) pressure levels. It appears that the limited vertical resolution of the latter is visible in some figures (e.g. fig. 2 and 3). The same may apply to GFS model versus pressure-level data.

Models do certainly suffer from too strong numerical diffusion. Nevertheless, did the authors verify that their use of pressure-level instead of model-level data does not have any impact on the comparisons in sections 3 and 4?

Page 12, figure 5: the caption should refer to the (central) wavelengths of the MTSAT-2 IR channel used here (10.8 and 6.75 $\mu$m).

Section 5: given the scatter plots in fig.7, the authors should have noticed that the models appear to simulate very dry situations with comparable frequencies, while the aircraft data tend to show very dry situations with a significantly higher frequency. The "RO relative humidity vs. CONTRAST" in fig.6 (bottom-right) shows a similar pattern. I do not assume that the authors could explain the reason for these (common) features, but they should at least mention this apparent discrepancy.

Page 13, figure 6: given the distribution of points in the scatter plots, and that refractivity primarily depends on density resp. pressure, did the authors consider to use a logarithmic scale for refractivity?

Section 7, conclusion 7: "When compared to CONTRAST, RO has a moist bias for low humidity values, and a dry bias for high humidity values". While this may be true for low

humidity values, I am less convinced by the results from section 5 that this is true for high humidity values, as the correlation fit is assumed to be valid for the full (log-scale) humidity range. Restricting the fit to e.g. the range q > 1 g/kg, there appears to be only a small bias.

Technical corrections:

Page 4, lines 6-7: The official spelling is "Metop", not "METOP". See EUMETSAT's web site or the WMO OSCAR database. Similarly "TerraSAR-X", not "TerrSarX".

References:

Pan et al.: journal reference?

Randel et al.: more details needed (journal reference?)

———————————————————

---

## Referee Comment (RC2) · Anonymous Referee #2 · 31 Oct 2016

Summary: This paper's objective is to assess the effectiveness of the GNSS radio occultation (RO) technique in characterizing extremely dry atmospheric layers. Comparisons are made during the CONTRAST campaign in Jan+Feb 2014, to other observations such as AIRS, MTSAT, aircraft and radiosonde profiles, and to reanalyses. The authors note the different sampling and resolution characteristics of these techniques and find good agreement generally between the different methods. The authors conclude that the GNSS RO technique is effective in contributing significant information about dry layers in the tropics and sub-tropics. A combination of case studies and statistics (for 2014) are used.

Review summary: The authors develop a useful study that is a significant contribution to the literature and should be published. However, certain conclusions contained within the paper are too narrowly drawn to be useful and should be modified or replaced. Further detailed comments follow.

Detailed Comments: Page 3, Line 1 (P3,L1): The reference to Randel 2016 is vague. Is this paper in preparation, submitted, etc.?

P 5, L 19: The instrument resolution is not the same as the retrieved product resolution for AIRS. The authors should quote the horizontal resolution of the retrieved temperature and humidity Level 2 products, not the instrument resolution.

P 5, L 28: I believe it is incorrect to suggest that the in-situ measurements have large sampling errors by comparison. Sampling error is the difference between a measurement value and the actual atmospheric state given the sampling volume.

P6, L11: This reads as follows: the time and space criteria consist of three pairs of values. What is meant by "closest" in time and space? Given the pairs, it is ambiguous. Which pair represents the maximal degree of collocation?

P6, L19: It's not clear that the GFS and ERA profile shapes are limited by "lower vertical resolution" (Figure 2). How many levels are there between 0 and 2.5 km? It may be more subtle factors such as model physics, or limited observations, that account for the profile shape.

P10, L7: I agree that AIRS vertical resolution cannot capture the transition, but the number of levels of GFS or ERA may be able. Please state the number of levels in these models for the different lower troposphere altitude ranges.

P11, L10: Please provide reference(s) to the previous work mentioned.

P11, L30: Given the similarities mentioned, is there a compelling reason not to show the aggregate statistics for all the collocation criteria?

P12, L5: Figure 6 as presented has limited value because the contribution of collocation error to RH scatter is completely unconstrained, thus compromising the value of this information in assessing RO. An attempt should be made to assess the contribution of collocation error (e.g. vary collocation criteria in time and space to assess scatter growth with collocation distance, or use reanalyses to assess – in a lower-bound sense – the contribution from collocation).

P15, L16: Appendix A has limited value, as it represents a single profile, but the difference between 1DVAR and "simple" depends critically on the accuracy of the temperature used in simple. Additional factors affect 1DVAR accuracy. A simulation or analytical treatment of the technique differences would be more useful, that spans a range of temperature errors (among other factors). It would seem more useful to quote the literature on errors of the simple method (e.g. Vergados et al. publications) and compare these errors to what is expected in the 1DVAR. I recommend removing or strongly modifying Appendix A to take these factors into account. If Appendix A is removed, the literature should be consulted as to how 1DVAR and simple might differ, and the appropriate references included and quoted.

P16, L3: I need to be convinced (based on the number of levels) that resolution is what limits GFS and ERA, versus other factors such as physics, or assimilation data set and method, etc.

---

## Author Comment (AC1) · 24 Nov 2016

We thank the anonymous referee for the thorough review and all comments. We will implement the following changes according to the referee's suggestions. We have answered all comments below (for easier comparison the referee comments are included in italic).

*Page 3, Line 1 (P3, L1): The reference to Randel 2016 is vague. Is this paper in preparation, submitted, etc.?*
The complete reference has been provided.

*P5, L19: The instrument resolution is not the same as the retrieved product resolution for AIRS. The authors should quote the horizontal resolution of the retrieved temperature and humidity Level 2 products, not the instrument resolution.*
We changed the sentence to:

"The AIRS Level 2 products are reported on 28 standard pressure levels between 1100 hPa and 0.1 hPa. They have a horizontal resolution of 50 km[1], and a vertical resolution of ∼1 km for temperature and ∼2 km for humidity[2]."

*P5, L28: I believe it is incorrect to suggest that the in-situ measurements have large sampling errors by comparison. Sampling error is the difference between a measurement value and the actual atmospheric state given the sampling volume.*
We should have used "representativeness error" or "representativeness difference" rather than "sampling error". We modified the sentence beginning with "As such....." in line 26 to:

"AIRS, RO, and models, on the other hand, represent averages over much larger horizontal and vertical scales of observation (larger horizontal and vertical footprints). Thus different volumes of air are sampled and compared, leading to representativeness errors or differences due to their different horizontal and vertical footprints, especially when measuring fields with high temporal and spatial variability such as water vapor and relative humidity."

*P6, L11: This reads as follows: the time and space criteria consist of three pairs of values. What is meant by "closest" in time and space? Given the pairs, it is ambiguous. Which pair represents the maximal degree of collocation?*
We clarified this discussion as follows:
* * *
[1] http://disc.gsfc.nasa.gov/uui/datasets/AIRS2RET_V006/summary
[2] http://airs.jpl.nasa.gov/data/physical_retrievals

"Note that the shortest time windows in these criteria correspond to the longest spatial intervals; this is done to ensure enough pairs in sets matching each of the criteria. Using a criterion of the shortest time and space separation would not yield enough pairs to make the results as meaningful. As shown in Section 5 below, all three of our criteria gave similar results; thus we only show results for the 3 h and 600 km criteria."

We also show the statistics showing the similar results (see response to the comment P11, L30 below).

*P6, L19: It's not clear that the GFS and ERA profile shapes are limited by "lower vertical resolution" (Figure 2). How many levels are there between 0 and 2.5 km? It may be more subtle factors such as model physics, or limited observations, that account for the profile shape.*
*P10, L7: I agree that AIRS vertical resolution cannot capture the transition, but the number of levels of GFS or ERA may be able. Please state the number of levels in these models for the different lower troposphere altitude ranges.*
*P16, L3: I need to be convinced (based on the number of levels) that resolution is what limits GFS and ERA, versus other factors such as physics, or assimilation data set and method, etc.*
Since both reviewers commented on the vertical resolution of ERA and GFS, we provide a response here adressing all concerns of both reviewers:

Vertical profiles from the ERA and GFS analyses interpolated to RO locations are provided by COSMIC CDAAC for pressure levels (while the RO physical profiles are given on a 100 m grid). GFS is given on the following pressure levels: from 1000 hPa to 250 hPa every 50 hPa, and additionally on 975 hPa and 925 hPa (plus additional levels above 250 hPa which are not relevant here). ERA is given on the following pressure levels: from 1000 hPa to 750 hPa at 25 hPa steps, and from 750 hPa to 250 hPa at 50 hPa

steps (plus additional levels above 250 hPa which are not relevant here). Assuming a scale height of 8 km, this yields a total of 18 levels for ERA and 15 levels for GFS where RO provides reliable moisture information (below 8 km, about 375 hPa). There are 80 RO levels between the surface and 8 km. Thus the vertical resolution of the ERA and GFS analyses provided by CDAAC is much lower than that of the RO observations.

CDAAC does not provide RO-collocated model profiles on model levels. Thus we downloaded an example day of the ERA fields for both pressure and model levels and converted model levels to pressure levels ourselves. Assuming a scale height of about 8 km, ERA provides 18 pressure levels below 8 km (corresponding to about 375 hPa) and 25 model levels. A few of these extra model levels are at very low altitudes (near the surface, at altitudes with pressure greater than 1000 hPa, which is the lowest given pressure level for ERA on a pressure grid). Consequently, this leaves only a few more extra model levels that would increase the vertical resolution when compared to pressure levels.

Assuming again a scale height of 8 km, a pressure of 750 hPa corresponds to 2.4 km. This yields 8 pressure levels for GFS and 11 pressure levels for the ERA between 1000 hPa and 750 hPa (surface to 2.4 km). The vertical separation between levels increases further when going to higher altitudes. The smoothness (lack of vertical detail) of the GFS and ERA profiles compared to the RO and CONTRAST profiles is an indication of the lower vertical resolution of these models compared to RO and CONTRAST. The important point is that the overall shape of the GFS and ERA profiles is similar to the overall shape of the higher-resolution profiles (RO and CONTRAST). Other factors such as model physics or limited observations could not increase the vertical resolution of the GFS and ERA profiles, but they could change the overall shape.

The bottom line: increasing the vertical resolution of the GFS and ERA profiles by adding a few more levels in the vertical would not change the overall shape nor the conclusions. We are confident in our comments on P6 L18 and P10 L8 that the less

sharp vertical gradients in moisture at the top of the moist layers in the model analyses are "partly due to lower vertical resolution".

*P11, L10: Please provide reference(s) to the previous work mentioned.*
The "narrow dry tongues" actually refer to stratospheric intrusions on the rear of strong extratropical cycles (as in Young et al. (1987), Browning and Dicks (2001), and Keyser and Shapiro (1986)) and not the large-scale sinking air that is studied here. Thus we removed the entire paragraph (P11, L9-11) and instead edited the paragraph starting at P10, L14:

"Fig. 4, top right, shows the ERA $RH$ fields at 500 hPa for the whole region. The large-scale region of very dry air extends from 110°E to 180°E with a width of 1600 km to 2200 km. It also shows the high horizontal variability of moisture."

*P11, L30: Given the similarities mentioned, is there a compelling reason not to show the aggregate statistics for all the collocation criteria?*
In the revised paper we now present the aggregate statistics for relative humidity differences between RO and CONTRAST, ERA and CONTRAST, and GFS and CONTRAST for the three collocation criteria. We added the following sentence at page 11, line 30 (after "..profile statistics and scatter plots"):

"To illustrate the similarities, we show the mean, root mean square (RMS), and Pearson R correlation coefficient for relative humidity $RH$ for RO-CONRAST, ERA-CONTRAST and GFS-CONTRAST differences for all three criteria in Table 2. We chose $RH$ due to our focus on tropospheric moisture in this paper. Furthermore, $RH$ is the only parameter without some inherent vertical correlation due to a general decrease with altitude (i.e. $N$, $T$, and $q$)."

Furthermore, we accidentally included data above 10 km in the scatter plots, but

**Table 2.** Mean, RMS, and Pearson R coefficients for differences in relative humidity $RH$ of RO-CONTRAST, ERA-CONTRAST, and GFS-CONTRAST for 3 different collocation criteria.

|  |  | 3 h 600 km
644 points | 12 h 300 km
687 points | 24 h 200 km
419 points |
|---|---|---|---|---|
| **RO-CONTRAST** | mean | -4.0 | -6.5 | -0.2 |
|  | RMS | 21.3 | 23.4 | 22.5 |
|  | Pearson R | 0.782 | 0.758 | 0.751 |
| **ERA-CONTRAST** | mean | -3.9 | -5.4 | 0.6 |
|  | RMS | 20.0 | 20.5 | 21.8 |
|  | Pearson R | 0.807 | 0.807 | 0.760 |
| **GFS-CONTRAST** | mean | -5.0 | -6.3 | -0.1 |
|  | RMS | 20.7 | 21.4 | 21.9 |
|  | Pearson R | 0.799 | 0.798 | 0.757 |

discarded some of the data below. In the revised paper all data points below 10 km are included in the scatter plots. This does not change the results or the conclusions we can draw from them. The corrected scatter plots are included in the revised manuscript.

*P12, L5: Figure 6 as presented has limited value because the contribution of collocation error to RH scatter is completely unconstrained, thus compromising the value of this information in assessing RO. An attempt should be made to assess the contribution of collocation error (e.g. vary collocation criteria in time and space to assess scatter growth with collocation distance, or use reanalyses to assess – in a lower-bound sense – the contribution from collocation).*
Comparisons using 8 years of radiosonde (2 stations) and RO data using very tight collocation criteria (1 h, 100 km) still showed highly scattered data for $RH$ (Figure 1). Thus the collocation errors are not the dominant factor in the large scatter between RO and CONTRAST relative humidities. Relative Humidity is a highly variable parameter, even if comparisons are made that close in time and location. Tighter criteria are

**Fig. 1.** Scatter plot of RO relative humidity and radisonde (RS) relative humidity for pairs of RO and RS data located within 1 h and 100 km.

maybe possible using a long radiosonde record, however, with the very limited number of collocations of CONTRAST and RO, that is not possible. The point of Fig. 6 lower right and Fig. 7 left is that $RH$ is a highly variable parameter and that small differences in temperature and/or specific humidity make large differences in relative humidity. These differences occur when the scale (footprint) of the different observations vary, as they do between RO and CONTRAST and ERA and CONTRAST. The scatter is less when the footprints of the two correlated $RH$ values are similar, as in GFS vs. ERA (Fig. 7, right), but the scatter is still large compared to the scatter of temperature, refractivity, and specific humidity (Fig. 6).

*P15, L16: Appendix A has limited value, as it represents a single profile, but the difference between 1DVAR and "simple" depends critically on the accuracy of the temperature used in simple. Additional factors affect 1DVAR accuracy. A simulation or analytical treatment of the technique differences would be more useful, that spans a range of temperature errors (among other factors). It would seem more useful to quote the literature on errors of the simple method (e.g. Vergados et al. publications) and compare these errors to what is expected in the 1DVAR. I recommend removing or strongly modifying Appendix A to take these factors into account. If Appendix A is removed, the literature should be consulted as to how 1DVAR and simple might differ, and the appropriate references included and quoted.*

We rewrote Appendix A and added figures that further explore the differences in moisture resulting from applying the simple retrieval versus the 1D-Var and included references to the work of Vergados and colleagues. Appendix A now contains statistics of many comparisons of the simple and 1D-Var retrievals in addition to the one example:

**Appendix A: RO Water Vapor Retrievals:**

The two common techniques to retrieve physical temperature and water vapor profiles from RO are the so-called simple retrieval (Kursinski and Hajj, 2001) and the 1D-Var retrieval. For the simple retrieval, $e$ is derived via Eq. 1 using the RO observed $N$, and $T$ from an independent source (e.g. radiosonde, model, or analysis). Advantages of the simple retrieval are its simplicity and ease of calculation, and its independence of model moisture (and thus independence from errors in model moisture). Vergados et al. (2015) used the simple method for this reason, using ECMWF temperatures for the independent temperatures. Ware et al. (1996) (Eq. 3 and 4 in their manuscript) noted that for a perfect $N$ and $p$, the error (difference) in $e$ related to an error (difference) in $T$ can be approximated by:

$$\Delta e \approx \frac{2TN - 77.6p}{3.73 \times 10^5} \times \Delta T A1 \tag{1}$$

The simple method provides good results ($\Delta e < 0.25$ hPa) in the lower troposphere if the ancillary temperature data are reasonably accurate ($\Delta T < 1$ K). In the 1D-Var procedure, a-priori (first guess or background) profiles of $T$ and $e$ are obtained from independent observations and adjusted toward the RO measurements by a statistical optimization procedure (Poli et al., 2002; COSMIC, 2005). The 1D-Var procedure considers the statistics of errors in the RO observations as well as the statistical errors in the a-priori information, to achieve a consistent temperature and water vapor profile that minimizes, in a statistical sense, the errors in $T$ and $e$.

Because both the 1D-Var and simple method are used in different studies to estimate water vapor, it is important to understand how the results from the two methods compare. In this Appendix we compare the two methods using the data in our study by first showing an example and then statistics using a large number of data pairs. Fig. 2 shows the parameters $RH$, $T$, $q$, and $e$ for the first guess (ERA, solid) and retrieved

**Fig. 2.** $T$, $RH$, $q$, and $e$ profiles for ERA (solid, a-priori), RO 1D-Var (dashed), and RO simple retrieval (dotted).

from RO with the 1D-Var (dashed) and the simple retrieval (dotted) for an example profile.

The $RH$ (top left) is very low between 4 km and 9 km. The simple retrieval and the 1D-Var agree very well up to 4 km. At 4.2 km and 6 km to 9 km, the simple retrieval produces a negative $RH$ (due to negative $e$ values). In the simple retrieval, any error in $T$ will produce an error in $e$, and for dry air ($e$ close to zero), this error may lead to an unphysical negative value for $e$, $q$, and $RH$. The 1D-Var can theoretically also reach negative values in these situations, but it is artificially set to a very small positive value ($10^{-6}$ hPa) in the COSMIC CDAAC 1D-Var retrieval.

The RO 1D-Var and first guess (ERA) $T$ (top right) agree very well; temperature differences are within 1.5 K throughout the profile. (The RO $T$ in the simple retrieval assumes ERA $T$ to be the truth, so it is identical to the ERA $T$ in this figure.)

The bottom panels show $q$ and $e$. Both parameters become negative above 4.2 km in the simple retrieval.

Generally, the moisture profiles derived from both the simple retrieval and the 1D-Var show much more vertical structure than the ERA profile; this structure comes from the vertical structure of the RO refractivity profile. The above example shows that the simple and 1D-Var methods give very similar results for temperature, specific humidity, water vapor pressure, and even relative humidity up to the bottom of the very dry layer (a little above 4 km) where the water vapor pressure becomes less than 0.1 hPa. The close agreement in this example is typical, as shown by a comparison of the simple and 1D-Var method over a large number of cases.

We compared values of $T$, $q$ and $RH$ computed from the two methods, using ERA

**Fig. 3.** Specific humidity from the RO 1D-Var retrieval (left) and simple retrieval (right) versus radiosonde data. Collocation criteria: 2 h 200 km, altitude range 1000 hPa to 200 hPa.

**Fig. 4.** Specific humidity from the RO 1D-Var retrieval compared to the RO simple retrieval, using the same profiles as in Fig 3. Points from profiles for which the collocated radiosonde profile experiences super-refraction are marked by an X.

$T$ directly for the simple retrieval and ERA $T$ and $q$ as the first guess for the 1D-Var estimates, against radiosonde data from Vacoas, Mauritus (20.3°S, 57.5°E), which is located in a region that frequently contains very dry layers. We used the period 2006 to 2014, using collocation criteria of 2 h and 200 km. Because of large uncertainties in radiosonde humidity measurements above 10 km (Miloshevich et al., 2006), we focus on comparisons over 1000 hPa to 200 hPa.

Scatter plots of $q$ from the 1D-Var retrieval (left) and the simple retrieval (right) versus the independent radiosonde observations are shown in Fig. 3. The results are very similar with a correlation of 0.914 for the 1D-Var retrieval and 0.908 for the simple method.

Fig. 4 uses the same data as above, but shows the scatter plot of specific humidities from the 1D-Var versus simple retrieval. The retrieved values from the two methods are very similar with a correlation of 0.994. Approximately 15 points (blue, i.e. pressure altitudes of about 800 hPa) show 1D-Var values that are significantly higher than the simple values. We suspected that these points were from profiles where super-refraction occurs. In the case of super-refraction, $N$ is biased negatively (Sokolovskiy, 2003) and $e$ from the simple retrieval, which uses an accurate estimate of $T$, will be too low. In 1D-Var, the negative RO $N$ bias will be mitigated to some extent and the resultant 1D-Var temperature will be too high and the water vapor pressure too low, but not as low as in the simple retrieval. Thus $e$ from the simple method will be significantly

lower than the $e$ from the 1D-Var method under conditions of super-refraction.

To test this hypothesis, we checked all radiosonde profiles in the pairs for the criterion for super-refraction ($dN/dz < -157$ N-units km$^{-1}$). If a profile contained this critical value at some pressure level, we marked all data points of that profile with an X in Fig. 4. Indeed, most of the points with strong differences between 1D-Var and simple appear to occur with super-refraction.

We note that neither the 1D-Var nor the simple method for computing water vapor pressure at high altitudes ($p <$ 200 hPa) from observed RO refractivity $N$ will provide reliable estimates, because at these altitudes water vapor contributes very little to the refractivity; i.e. the so-called "wet" term in Eq. 1 is less than 1% of the first, or "dry" term as noted by Wang et al. (2013). There is simply not enough information on water vapor pressure in refractivity at these altitudes to retrieve accurate estimates of water vapor. For example, the mean hurricane-season tropical atmosphere (Dunion and Marron, 2008) gives the following value of $T$ at 200 hPa (about 12.4 km altitude): $T = -54.6\,°\mathrm{C}$ (218.6 K). The saturation vapor pressure at this temperature is 0.04 hPa; thus for 100% relative humidity, the dry term for $N$ is 71.0 and the wet term is 0.31, or 0.4% of the refractivity value. For the above values, the relationship between errors in $e$ and $T$ (Eq. 1) gives $\Delta e = 0.042\Delta T$ (hPa). So a temperature error (difference) of 0.5 K gives a difference in $e$ of 0.021 hPa, which is more than 50% of the saturation vapor pressure at this temperature. This example illustrates the difficulty in calculating water vapor pressure and relative humidity in the upper troposphere. Vergados et al. (2014) estimated the retrieval errors in specific humidity at different pressure levels (925, 850, 700, 500, and 400 hPa). They estimated that in the lower troposphere (925, 850 and 700 hPa) the percentage error in specific humidity for a temperature uncertainty of 1 K was less than 3 % in the tropics, 6 % in middle latitudes, and 10 % in high latitudes. At 400 hPa, the percent errors grew to 18 % in the tropics, 45,% in middle latitudes, and 67 % in high latitudes.

**References**

Browning, K. A. and Dicks, E. M.: Mesoscale structure of a polar low with strong upper-level forcing, Quart. J. Roy. Meteor. Soc., 127, 359–375, 2001.

COSMIC: Variational Atmospheric Retrieval Scheme (VARS) for GPS Radio Occultation Data, COSMIC technical report, version 1.1, University Corporation for Atmospheric Research, 2005.

Dunion, J. P. and Marron, C. S.: A Reexamination of the Jordan Mean Tropical Sounding Based on Awareness of the Saharan Air Layer: Results from 2002, J. Climate, 21, 5242–5253, 2008.

Keyser, D. and Shapiro, M. A.: A Review of the Structure and Dynamics of Upper-Level Frontal Zones, Mon. Wea. Rev., 114, 452–499, 1986.

Kursinski, E. R. and Hajj, G. A.: A comparison of water vapor derived from GPS occultations and global weather analyses, J. Geophys. Res., 106, 1113–1138, 2001.

Miloshevich, L. M., Vömel, H., Whiteman, D. N., Lesht, B. M., Schmidlin, F. J., and Russo, F.: Absolute accuracy of water vapor measurements from six operational radiosonde types launched during AWEX-G and implications for AIRS validation, J. Geophys. Res., 111, 2006.

Poli, P., Joiner, J., and Kursinski, E.: 1DVAR analysis of temperature and humidity using GPS radio occultation refractivity data, J. Geophys. Res., 107, 2002.

Sokolovskiy, S.: Effect of superrefraction on inversions of radio occultation signals in the lower troposphere, Radio Sci., 38, 2003.

Vergados, P., Mannucci, A. J., and Ao, C. O.: Assessing the performance of GPS radio occultation measurements in retrieving tropospheric humidity in cloudiness: A comparison study with radiosondes, ERA-Interim, and AIRS data sets, J. Geophys. Res., 119, 7718–7731, 2014.

Vergados, P., Mannucci, A. J., Ao, C. O., Jiang, J., and Su, H.: On the comparisons of tropical relative humidity in the lower and middle troposphere among COSMIC radio occultations and MERRA and ECMWF data sets, Atmos. Meas. Tech., 8, 1789–1797, 2015.

Wang, J., Zhang, L., Lin, P.-H., Bradford, M., Cole, H., Fox, J., Hock, T., Lauritsen, D., Loehrer, S., Martin, C., VanAndel, J., Weng, C.-H., and Young, K.: Water vapor variability and comparisons in subtropical Pacific from T-PARC Driftsonde, COSMIC, and reanalyses, J. Geophys. Res., 115, 2010.

Wang, B.-R., Liu, X.-Y., and Wang, J.-K.: Assessment of COSMIC radio occultation retrieval

product using global radiosonde data, Atmos. Meas. Tech., 6, 1073–1083, 2013.

Ware, R., Exner, M., Gorbunov, M., Hardy, K., Herman, B., Kuo, Y., Meehan, T., Melbourne, W., Rocken, C., Schreiner, W., Sokolovskiy, S., Solheim, F., Zou, X., Anthes, R., Businger, S., and Trenberth, K.: GPS Sounding of the Atmosphere from Low Earth Orbit: Preliminary Results, Bull. Amer. Meteor. Soc., 77, 1996.

Young, M. V., Monk, G. A., and Browning, K. A.: Interpretation of Satellite Imagery of A Rapidly Deepening Cyclone, Quart. J. Roy. Meteor. Soc. 113 (478), 1089–1115, 1987.

[Figure]

**Fig. 5.**

[Figure]

**Fig. 6.**

[Figure]

**Fig. 7.**

[Figure]

**Fig. 8.**

---

## Author Comment (AC2) · 24 Nov 2016

We thank the anonymous referee for the review and the helpful and constructive comments. We will implement the following changes according to the referee's suggestions. We have answered all comments below (for easier comparison the referee comments are included in italic).

*Page 2, lines 2-3: In my opinion there is no doubt in the literature that super-refraction conditions do exist. I therefore recommend to reformulate: "if super-refraction conditions exist" → "under super-refraction conditions".*
Thank you, we corrected the sentence to make a clearer statement.

[Figure]

*Section 2.2 ("The CONTRAST experiment"): For the reader's sake, there should be a very brief explanation of the sensors used for the aircraft observations, in particular the derivation of water vapor pressure, as this is a central quantity in this study. The Pan et al. paper has no full reference and could not be found.*
We added the complete Pan et al. (2016) reference (see below) and the following paragraphs:

"Water Vapor was measured by the Vertical Cavity Surface Emitting Laser (VCSEL) hygrometer (absolute concentration of water vapor in molecules per cubic cm). It is designed to work throughout the troposphere (and also the lower stratosphere), and has an accuracy of $\pm 6\,\%$ mixing ratio +0.3 ppmv and a precision of $\leq 3\,\%$ (see Zondlo et al. (2010) for details). Temperature was measured by two Harco heated total air temperature sensors (estimated accuracy: $0.5\,°C$, precision: $< 0.01\,°C$), pressure was measured using the Parascientific Sensor, Model 1000 transducer (accuracy: 0.1 hPa, precision: $<0.01$ hPa)[1]. From the CONTRAST netcdf files, the variables used for $T$, $e$, and $p$ are ATX, EW_VXL, and PSXC."

*Section 2.3 ("ERA-Interim Reanalysis") and section 2.4: The ERA-Interim dataset is available on the original 60 model levels, as well as interpolated to a predefined set of (coarser) pressure levels. It appears that the limited vertical resolution of the latter is visible in some figures (e.g. fig. 2 and 3). The same may apply to GFS model versus pressure-level data. Models do certainly suffer from too strong numerical diffusion. Nevertheless, did the authors verify that their use of pressure-level instead of model-level data does not have any impact on the comparisons in sections 3 and 4?*
Since both reviewers commented on the vertical resolution of ERA and GFS, we provide a response here adressing all concerns of both reviewers:

[Figure]
* * *
[1]https://www2.acom.ucar.edu/sites/default/files/seac4rs/StateParameters.pdf

Vertical profiles from the ERA and GFS analyses interpolated to RO locations are provided by COSMIC CDAAC for pressure levels (while the RO physical profiles are given on a 100 m grid). GFS is given on the following pressure levels: from 1000 hPa to 250 hPa every 50 hPa, and additionally on 975 hPa and 925 hPa (plus additional levels above 250 hPa which are not relevant here). ERA is given on the following pressure levels: from 1000 hPa to 750 hPa at 25 hPa steps, and from 750 hPa to 250 hPa at 50 hPa steps (plus additional levels above 250 hPa which are not relevant here). Assuming a scale height of 8 km, this yields a total of 18 levels for ERA and 15 levels for GFS where RO provides reliable moisture information (below 8 km, about 375 hPa). There are 80 RO levels between the surface and 8 km. Thus the vertical resolution of the ERA and GFS analyses provided by CDAAC is much lower than that of the RO observations.

CDAAC does not provide RO-collocated model profiles on model levels. Thus we downloaded an example day of the ERA fields for both pressure and model levels and converted model levels to pressure levels ourselves. Assuming a scale height of about 8 km, ERA provides 18 pressure levels below 8 km (corresponding to about 375 hPa) and 25 model levels. A few of these extra model levels are at very low altitudes (near the surface, at altitudes with pressure greater than 1000 hPa, which is the lowest given pressure level for ERA on a pressure grid). Consequently, this leaves only a few more extra model levels that would increase the vertical resolution when compared to pressure levels.

Assuming again a scale height of 8 km, a pressure of 750 hPa corresponds to 2.4 km. This yields 8 pressure levels for GFS and 11 pressure levels for the ERA between 1000 hPa and 750 hPa (surface to 2.4 km). The vertical separation between levels increases further when going to higher altitudes. The smoothness (lack of vertical detail) of the GFS and ERA profiles compared to the RO and CONTRAST profiles is an indication of the lower vertical resolution of these models compared to RO and CONTRAST. The important point is that the overall shape of the GFS and ERA profiles

is similar to the overall shape of the higher-resolution profiles (RO and CONTRAST). Other factors such as model physics or limited observations could not increase the vertical resolution of the GFS and ERA profiles, but they could change the overall shape.

The bottom line: increasing the vertical resolution of the GFS and ERA profiles by adding a few more levels in the vertical would not change the overall shape nor the conclusions. We are confident in our comments on P6 L18 and P10 L8 that the less sharp vertical gradients in moisture at the top of the moist layers in the model analyses are "partly due to lower vertical resolution".

*Page 12, figure 5: the caption should refer to the (central) wavelengths of the MTSAT-2 IR channel used here (10.8 µm and 6.75 µm).*
We added the wavelengths of the used MTSAT-2 channels to the figure caption.

*Section 5: given the scatter plots in fig. 7, the authors should have noticed that the models appear to simulate very dry situations with comparable frequencies, while the aircraft data tend to show very dry situations with a significantly higher frequency. The "RO relative humidity vs. CONTRAST" in fig. 6 (bottom-right) shows a similar pattern. I do not assume that the authors could explain the reason for these (common) features, but they should at least mention this apparent discrepancy.*
Section 5, Fig.7: We changed the discussion about $RH$ in Figs. 6 and 7 on page 12 lines 16-17 to:

"$RH$ plots (bottom right) are highly scattered and have a lower correlation coefficient of around 0.78 with a bias and large spread in the data sets. The moist bias of RO for very dry air was already noted in the paragraph above. Thus CONTRAST shows a much higher frequency of very low $RH$ values than both RO (Fig. 6 lower right) and ERA (Fig. 7, left). The large spread can be explained by several factors: 1) $RH$ is

sensitive to both small variations in $T$ and $q$, thus representativeness differences or errors of both $T$ and $q$ contribute to differences in $RH$; 2) $RH$ does not have a vertical profile with a mean structural or climatological variation in the vertical as $N$, $T$, $q$ do (with an overall decrease with altitude); and 3) $RH$ can undergo extremely strong changes in the vertical..”

We added the following paragraph to the discussion of $q$ (page 12, line 15):

“Because of the highly accurate aircraft water vapor and temperature measurements and the very small scale of the observation (essentially a point observation), the CONTRAST measurements are capable of detecting extremes of dry and moist air more frequently than RO observations or model estimates, whose data represent averages over larger scales.”

*Page 13, figure 6: given the distribution of points in the scatter plots, and that refractivity primarily depends on density resp. pressure, did the authors consider to use a logarithmic scale for refractivity?*
We have also made the same plot using logarithmic scales, but because $N$ has much less variability than $q$, using logarithmic scales does not reveal any additional features.

*Section 7, conclusion 7: “When compared to CONTRAST, RO has a moist bias for low humidity values, and a dry bias for high humidity values”. While this may be true for low humidity values, I am less convinced by the results from section 5 that this is true for high humidity values, as the correlation fit is assumed to be valid for the full (log-scale) humidity range. Restricting the fit to e.g. the range $q < 1$ g/kg, there appears to be only a small bias.*
The dry bias is difficult to see in Fig. 6 lower left, but it is there. Plotting on a linear-linear scale rather than log-log scale clearly shows a dry bias of RO for higher humidity values (see Fig. 1).

**Fig. 1.** Same as Fig. 6 lower left in the manuscript, but plotted on a linear-linear scale.

*Technical corrections: Page 4, lines 6-7: The official spelling is "Metop", not "METOP".
See EUMETSAT's web site or the WMO OSCAR database. Similarly "TerraSAR-X",
not "TerrSarX".*
Thank you, we corrected the spelling of Metop and TerraSAR-X.

*References: Pan et al.: journal reference? Randel et al.: more details needed (journal
ref)*
Pan et al.: journal reference: BAMS, included now; Randel et al.: journal reference:
JGR, included now.

**References**

Pan, L. L., Atlas, E. L., Salawitch, R. J., Honomichl, S. B., Bresch, J. F., Randel, W. J., Apel, E. C., Hornbrook, R. S., Weinheimer, A. J., Anderson, D. C., Andrews, S. J., Baidar, S., Beaton, S. P., Campos, T. L., Carpenter, L. J., Chen, D., Dix, B., Donets, V., Hall, S. R., Hanisco, T. F., Homeyer, C. R., Huey, L. G., Jensen, J. B., Kaser, L., Kinnison, D. E., Koenig, T. K., Lamarque, J.-F., Liu, C., Luo, J., Luo, Z. J., Montzka, D. D., Nicely, J. M., Pierce, R. B., Riemer, D. D., Robinson, T., Romashkin, P., Saiz-Lopez, A., Schauffler, S., Shieh, O., Stell, M. H., Vaughan, G., Ullmann, K., Volkamer, R., and Wolfe, G.: The Convective Transport of Active Species in the Tropics (CONTRAST) Experiment, BAMS (in press), 2016.
Randel, W. J., Rivoire, L., Pan, L., and Honomichl, S.: Dry layers in the tropical troposphere observed during CONTRAST and global behavior from GFS analyses, JGR, submitted, 2016.
Zondlo, M. A., Paige, M. E., Massick, S. M., and Silver, J. A.: Vertical cavity laser hygrometer for the National Science Foundation Gulfstream-V aircraft, J. Geophys. Res., 115, 2010.

[Figure]

644 data points, mean=0.2, med=-0.0,
Pearson r=0.946,  RMS=1.6

[Figure]

Fig. 2.